

# Electromagnetic and DC-current geophysics for soil compaction assessment

Alberto Carrera[1], Luca Peruzzo[2], Matteo Longo[1], Giorgio Cassiani[2], Francesco Morari[1]

[1] DAFNAE, University of Padova, viale dell'Università 16, 35131 Legnaro, Italy

[2] Department of Geosciences, University of Padova, via Gradenigo 6, 35131 Padova, Italy

*Correspondence to*: Alberto Carrera (alberto.carrera@phd.unipd.it)

**Abstract.** Monitoring soil structure is of paramount importance due to its key role in the critical zone as the foundation of terrestrial life. Variations in the arrangement of soil components significantly influence its hydro-mechanical properties, and

therefore its impact on the surrounding ecosystem. In this context, soil compaction resulting from inappropriate agricultural practices not only affects soil ecological functions, but also decreases the water-use efficiency of plants by reducing porosity and increasing water loss through superficial runoff and enhanced evaporation.

In this study, we compared the ability of electric and electromagnetic geophysical methods, i.e. Electrical Resistivity Tomography and Frequency-domain Electromagnetic Method, to assess the effects of compaction on agricultural soil. The

objective was to highlight the electro-magnetic response caused by both heavy plastic soil deformations generated by a super-heavy vehicle and the more common tractor tramlines.

DC-current prospecting has finer spatial resolution and allows a tomographic approach, requiring higher logistic demands and the need for ground galvanic contact. On the other hand, contactless electromagnetic induction methods can be quickly used to define the distribution of electrical conductivity in the shallow subsoil in an easier way. Results, validated with

traditional soil characterization techniques (i.e. penetration resistance, bulk density and volumetric water content on collected samples), show the pros & cons of both techniques and how differences in their spatial resolution heavily influence the ability to characterize compacted areas with good confidence. This work aims at contributing to the methodological optimization of agro-geophysical acquisitions and data processing, in order to obtain accurate soil models through a non-invasive approach.


## 1 Introduction

Soil is the foundation of terrestrial life, and its structure and dynamics result from the intertwining of biotic and abiotic factors, as well as the preponderance of human action. Recent developments in sensing technology, analysis methods and data interpretation have paved the way for innovative approaches aimed at characterizing and safeguarding a wide spectrum

of soil-based ecosystem services. Over the past decades, digital soil mapping has emerged as a transformative approach in soil science (McBratney et al., 2003), with the goal of enhancing our understanding of soil properties, agricultural processes,



and moisture dynamics. The accessibility and affordability of ground-based and aerial sensor instruments have markedly improved, bringing high-resolution spatial-temporal data in support to traditional labor-intensive sampling techniques. Proximal and remote sensing techniques commonly rely on the use of instruments which can measure different portions of

the electromagnetic spectrum, improving the understanding of processes governing the soil–plant–atmosphere continuum (Viscarra Rossel et al., 2011; Mulder et al., 2011). Software and hardware, mostly of a non-invasive nature, are continuously optimized for agronomic applications, and also progressively deployed through airborne and unmanned vehicles (von Hebel et al., 2021).

Applied geophysics plays a key role in this context, and its use has become increasingly assiduous (Romero-Ruiz et al.,

2018; Garré et al., 2021). The preponderant methods employed here are based on the electrical properties of soil materials, which manifest main concomitant variations as the volumetric content and salinity of porous fluids change (Vereecken et al., 2007; Binley et al., 2015).

The frequency domain electromagnetic method (FDEM) can be considered the cornerstone of electromagnetic sensors optimized for soil applications due to their fast logistics and user-friendliness (Doolittle and Brevik, 2014; Hanssens et al.,

2019). By inducing electromagnetic fields underground and analyzing their interaction with the soil, this technique allows the electrical conductivity of large areas to be mapped non-invasively in short times (Boaga, 2017). This approach, especially if repeated over time, offers multifaceted advantages in the agronomic world, empowering farmers with information critical for precision agriculture practices (Corwin and Lesch, 2003; Lück et al., 2009). High-resolution soil variability and moisture EC-derived content and dynamics across the field, allows to implement precision irrigation

strategies, tailoring water application to the specific moisture needs of different areas (Fortes et al., 2015; Serrano et al., 2020). However, while its application is relatively simple and quick, FDEM method suffers from lower resolution than an in-situ tomographic approach (e.g., Electrical Resistivity Tomography, ERT) (Lavoué et al., 2010; Von Hebel et al., 2014; Busato et al., 2019; Bernatek-Jakiel and Kondracka, 2022), and therefore detailed and more imperceptible spatial heterogeneities can escape at both areal and especially depth scale.

ERT has become a ubiquitous instrument in agricultural research due to its inherent robustness and demonstrated adaptability across a spectrum of applications and spatial scales (Garré et al., 2012; Cassiani et al., 2015; Mary et al., 2018; Blanchy et al., 2020b; Carrera et al., 2022). Surveys are conducted through multi-electrode devices to capture the spatial distribution of electrical resistivity in the subsurface, thereby facilitating the generation of comprehensive 2D and 3D models.

Soil electrical conductivity (EC) is often used in the characterization of soil properties such as texture (Morari et al., 2009; Hanssens et al., 2019; Hubbard et al., 2021), soil nutrients and organic content (Heiniger et al., 2003; Martinez et al., 2009), but also to direct targeted sampling for detailed studies (Longo et al., 2020). Correlations between EC and soil properties, such as bulk density, porosity, and shear strength, are also used to identify soil compaction at the laboratory (Seladji et al., 2010) and field scales (Pentoś et al., 2021; Ren et al., 2022).



Soil compaction is a tangible manifestation of soil degradation. Heavy field traffic breakdowns soil aggregates altering the structure, limiting water and air infiltration and reducing root penetration (Berisso et al., 2012; Schjønning et al., 2019). In recent years, modern agricultural machinery has increased considerably in size, and with it, so has the compaction phenomenon (Raper, 2005; Nawaz F. et al., 2012). Its side effects have a significant impact on the soil ecosystem, particularly on hydrological regulation (i.e. surface runoff and reduced infiltration) and agronomic production (i.e. decreased

yields), resulting in significant ecological and economic damage to the entire society (Bronick and Lal, 2004; Hamza and Anderson, 2005). Therefore, a correct understanding of the processes involved in soil compaction, its identification and characterization, are necessary for prevention and to address future global challenges of sustainability and food security. This raises questions about the ability of geophysical methods to quantify the soil structure dynamics – including compaction – over space and time. Soil compaction exhibits highly spatial-temporal variability, depending on factors such as intensity and

distribution of machinery traffic and/or the implementation of tillage practices (Alaoui and Diserens, 2018; Piccoli et al., 2022). Consequently, studying and mapping soil compaction with geophysical techniques also remains a challenge. Field evidence (Garciá-Tomillo et al., 2018; Mansourian et al., 2023) and modelling approaches (Romero-Ruiz et al., 2022) identify compaction signature with increased electrical conductivity. However, there is little exploration of resolution and sensitivity aspects of the techniques used, which in fact form the basis of all subsequent studies dealing with the acquired

data.

In this work, we present the application of electromagnetic (frequency domain – or FDEM) and DC-current (Electrical Resistivity Tomography, or ERT) geophysics, quantitatively integrated with traditional soil characterization techniques (i.e. penetration resistance, bulk density and volumetric water content on collected samples) for the assessment of soil surface compaction. The survey was conducted both at the field scale, covering an area of 1.5 hectares, and in detail on individual

targeted transects. This combination of measurements explores the importance of the survey design on the sensitivity of the method to soil compaction, as well as the 2-D and 3-D spatial heterogeneity that is often difficult to image using punctual information only. The study aims at (a) comparing ERT and FDEM ability to identify soil compaction, (b) quantifying the degree of compaction and its hydro-geophysical consequences caused by a common tractor and a hyper-heavy machine in a typical clayey soil, and (c) contributing towards the methodological optimization of agro-geophysical acquisitions and data

processing, in order to obtain accurate soil models through a non-invasive approach. Results, validated with direct information, show the pros & cons of both FDEM and ERT techniques and how differences in their spatial resolution heavily influence their ability to characterize compacted areas with good confidence.

## 2 Material and Methods

### 2.1 Site description

The experiment was conducted in 2021 at the University of Padova's Experimental Farm "L. Toniolo," located in Legnaro, North-Eastern Italy (45°21′ N; 11°57′ E) (Fig. 1a). The investigation area presents a Fluvi-Calcaric Cambisol soil type





(WRB, 2014), characterized as a silt-loam soil with poor stratification and modest inherent fertility due to its limited organic carbon content (about 8-10 g kg$^{-1}$ within the 0 – 0.2 m layer, declining to 0.5 g kg$^{-1}$ at 0.6 - 0.9 m) and low cation exchange capacity (< 20 cmol kg$^{-1}$).

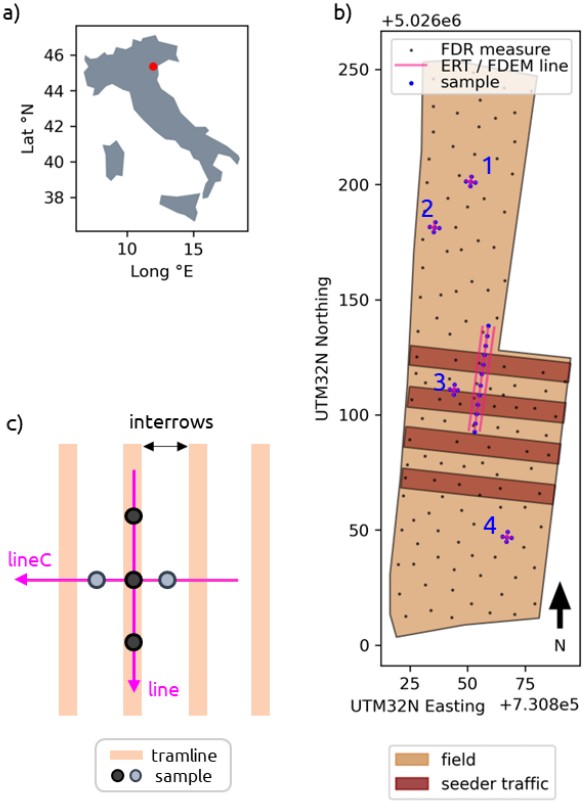


**Figure 1: a) site location and b) experimental field with traffic pattern and survey design. In the zoomed inset, c), transects orientation and samples position.**

The experiment was carried out in a bare soil area that is 60 m wide and 240 m long, for a total of 1500 m$^2$ (Fig. 1b). In the

past 60-yr the field was used for arable cropping following conventional agricultural practices, which involved moldboard plough 0.3-m depth and disk harrowing before seeding. Main crops were (*Zea mays L.*), winter wheat (*Triticum aestivum* L.), soybean (*Glycine max* (L.) Merr.), sugarbeet (*Beta vulgaris* L.), Italian ryegrass (*Lolium multiflorum* Lam.), etc. The soil was tilled as described above and then seeded (22$^{nd}$ & 23$^{rd}$ April 2021), either conventionally or by precision seeding. The vehicle used for common field works was a Fiat 680H of 2.8 tons, while precision seeding trial was performed with a

Lemken Azurit 10 seed drill of 1.5 ton mounted on a Fendt 718 tractor of around 8 tons with frontally attached a localized fertilization system, totaling about 11 tons. Each precision sowing event was performed with a single passage through the sowing steps highlighted in Fig. 1b. Data acquisition (fall season 2021) is described in detail in the following sections.



## 2.2 Frequency-domain Electromagnetic Method (FDEM)

Frequency-domain electromagnetic technique utilize**s** time variations in electromagnetic fields at relatively low frequencies (~1–100 kHz) and its functioning principle relies upon classical electromagnetic induction theory (McNeill, 1980; Deidda et al., 2014). Electro-Magnetic Induction (EMI) instruments measure the interaction between an induced primary electromagnetic field and the resultant secondary electromagnetic field. During the measurement process, the transmitter coil emits a primary time-varying electromagnetic field (Hp) that induces eddy currents increasing with increasing ground's

electrical conductivity (EC). This complex network of eddy currents induces a secondary electromagnetic field (Hs), which is jointly sensed by the receiver coils. From these measurements, an apparent electrical conductivity ($EC_a$) can be derived. Raw electrical conductivity values acquired through EMI surveys are "apparent" since they represent integrated values over depth. By varying the coil spacing or orientation, various subsurface depths can be probed (Blanchy et al., 2024). Inverse methods need to be used to convert the $EC_a$ (as a function of either frequency or coil setup) to a depth profile of true EC

(Von Hebel et al., 2019; McLachlan et al., 2021).

In this study, we adopted a CMD Mini-Explorer (GF Instruments), which contains three receiver coils with transmitter-receiver separation distances of 0.32m, 0.71m, 1.18m. The CMD was used in horizontal co-planar (HCP) and vertical co-planar (VCP) orientation, with respect to the ground, meaning that in total six depth-averaged readings were obtained for each measurement point (corresponding to the center of the instrument).

For an extensive survey, the device was pulled by a tractor using a 4m long rope, placing the instrument on a dedicated wood sledge at the soil surface. In this manner, no interaction with metallic (conductive) parts of the tractor or the sledge was ensured. The travel speed was approximately 6 km h$^{-1}$, with 0.5s of sampling rate, ensuring a spatial sampling density of approximately 0.8 m. The parallel transects were set about 4 m apart from each other, covering an area of about 1.5 hectares. For the high-resolution 2D transects, the device was hand-carried through its holding system at the soil surface, and the

travel speed was that of a slow walk (approximately 3.5 km h$^{-1}$). Measurements were logged every 0.5 s and paired with coordinates obtained from ProXT GPS receiver (Trimble, USA). In this case, spatial sampling was approximately 0.5 m.

The measured data were filtered to remove outliers (values outside the mean ± 2 standard deviations). Furthermore, a smoothing window was applied, replacing each data point with the average of its neighbors (number = 5) to favor a smoother inversion process. Finally, to define the maximum depth of the models, sensitivity profiles for each survey have been

calculated: all approximate zero toward 1.4 m below the surface. We set soil profiles composed of 24 layers with a thickness of 0.05 m each, and initial EC of 10 mS/m. Afterwards, the datasets were inverted using EMagPy (McLachlan et al., 2021), with the Full Maxwell (FS) forward model (Wait, 1982) and the Gauss-Newton optimization method (McLachlan et al., 2021) in order to minimize the total misfit between observed values and predicted values from the forward model solution. The choice of the FS forward operator, instead of the more frequently adopted Cumulative Sensity (CS), allows for the

calculation of a non-simplified response of the ground. EMagPy has the capability to perform quasi-2D inversions,





generating inverted EC depth profiles for each point of measurement, holding an average final Root Mean Square Percentage Error (RMSPE) of 6.1% for the 8 cross-transects and of 9.7% for the 3 longer lines.

## 2.3 Electrical Resistivity Tomography (ERT)

Electrical resistivity tomography is a well-established imaging technique and also long used in soil science (e.g. Samouëlian et al., 2005). ERT exploits multiple electrodes to measure the distribution of the electrical resistivity of the subsurface. Surveys are conducted through a quadrupole electrode arrangement: current is injected between a pair of electrodes, and the difference in electrical potential is measured between the other pair. From each measurement, an apparent resistivity value is derived, representing the equivalent resistivity of a homogenous subsurface.  Given multiple combinations of current and

potential electrodes along a transect, inverse modelling is then used to reconstruct a two- or three- dimensional image of the actual resistivity (Binley, 2015).

In this study, surveys were collected using a Syscal Pro 72 resistivimeter (Iris Instruments, Orleans, France) with an optimized dipole-dipole skip 0 scheme applied at 24 surface electrodes, acquiring both direct and reciprocal measurements, i.e. exchanging current and potentiometric electrodes for each quadrupole of measurement to get a statistical estimate of the

experimental error (Binley et al., 1995; Cassiani et al., 2006). Two sets of ERT transects were acquired at different scales to enhance the comparison with the EM data (Fig. 1). A first set of 3 longer lines (2 m electrodes spacing, above the seeder passages) was acquired to compare against plot-scale FDEM mapping. The second set of 8 short cross-transects (0.25 m electrodes spacing) was acquired to match and compare with the high-resolution EMI transects. Stainless steel electrodes were hammered into the first few centimeters of the soil to achieve the best compromise that would ensure electrical contact

and still abide by the general assumption of point-current injection. Contact resistances were checked before each acquisition, with very satisfactory values in the range $10^{-1}$ - $10^{0}$ kΩ.

Along each line, 477 quadrupoles were acquired, adopting a current injection time of 250 ms per cycle, with min and max stack numbers (number of cycles per quadrupole) of 3 and 6, and a quality factor (acceptable difference between cycle results) Q=2%. With these parameters, the total acquisition time for each line lasted around 8 min.

Datasets were analyzed in terms of direct-reciprocal deviation, discharging the quadrupoles with discrepancy larger than 5%, thus losing only a few dozen quadrupoles per line. The inversion process of the acquired datasets was performed adopting the same error threshold within the ResIPy software (Blanchy et al., 2020a), based on the R2/R3t codes based on an Occam's inversion method (Binley, 2015). All models converged within a maximum of 2 iterations, with a final RMS misfit of 1.0 each, thus confirming the excellent quality of data.



### 2.4 Soil sampling, penetration resistance and TDR measurements

Both geo-electric and electromagnetic surveys were acquired with the intent to map but also to characterize in detail portions of the field: an initial areal FDEM acquisition was followed by 3 additional lines to intercept seeder heavy passages, and 8 detailed transects, both FDEM and ERT (4 along and 4 across normal tractor tramlines) (Fig. 1c).

Survey positions were identified according to the FDEM spatial variability of the soil. The areal FDEM survey was used to provide ancillary data that help identify homogenous areas. Transect data were spatially interpolated using an Ordinary Kriging approach, and a k-means clustering algorithm was used to identify homogenous areas on the FDEM kriged maps. The k-means algorithm divides M points in N dimensions into K clusters to minimize the within-cluster sum of squares (Hartigan and Wong, 1979). Both spatial interpolation and cluster analysis were performed using ArcGIS Pro (ESRI, Redlands, CA). The objective function is calculated using Eq. (1) (Gore, 2008):

$$O_{KM} = \sum_{i=1}^{N} \sum_{h=1}^{H} d_{ih}^2 \tag{1}$$

where $d_{ij}$ is the component of a distance matrix, obtained from Eq. (2):

$$d_{ih}^2 = (x_i - c_h)' \boldsymbol{C^{-1}} (x_i - c_h) \tag{2}$$

where $c_h$ is the centroid of class $h$ and $C^{-1}$ is the inverse of covariance matrix of the independent variables, called the Mahalanobis distance (Varmuza and Filzmoser, 2016). The use of the Mahalanobis distance is justified by the fact that the six FDEM variables are correlated. The number of homogeneous areas were automatically selected resulting in four clusters (Fig. 2). For each area, one geophysical detailed survey (i.e. ERT + FDEM) was performed. Pairs of profiles were acquired, specifically: four "line" superimposed on the tramlines and four "lineC" crossing them orthogonally (Fig. 1c).

On top of each geophysical transect, three equally-spaced penetration resistance (PR) sampling zones were selected, for a total of 24 in-depth profiles. PR was measured at such points using a hydraulic-driven penetrologger (Eijkelkamp, Netherland), throughout the 0–80 cm soil layer, with a 30°, 2 cm$^2$ cone. Undisturbed 7.2 cm diameter soil samples were collected down to 0.7 m at the corresponding PR locations (24 in total) using a hydraulic sampler. Undisturbed soil samples were weighed, and a fraction (two-thirds) was oven-dried at 105 °C for 24 hours, to compute the gravimetric water content and bulk density (BD). The remaining soil fraction (one-third) was air-dried and sieved at 2-mm for texture analysis. Soil bulk density was estimated using the core method (Grossman and Reinsch, 2018), while soil texture was determined by laser diffraction (Mastersizer 2000; Malvern Panalytical Ltd, Malvern, UK), as described e.g. in Bittelli et al. (2019). In addition, along the longer ERT midline, 12 S PR depth profiles were acquired every 4m. During the same day and just after the geophysical survey, the volumetric surface water content in the entire field was measured using 128 geolocated spatially-distributed TDR measurements (FieldScout TDR 350 Soil Moisture Meter, Spectrum Technologies, Inc.) equipped with 22cm-long spikes.





 **2.5 Statistical analysis**

The level of dependency between soil electrical properties (i.e. $EC_{FDEM}$, $EC_{ERT}$) and basic physical soil properties (i.e. soil moisture, texture, bulk density, and penetration resistance) was calculated by the non-parametric Spearman's coefficients ($r_s$). Indeed, EC soil properties showed a non-normal distribution. Depths down to 0.3 m of the simultaneously collected parameters were considered, given the predominant compaction of interest observed in the shallowest layer and a general homogenization of trends at depth. The statistical analysis was performed using a Python routine based on SciPy (Virtanen et al., 2020).

# 3 Results

## 3.1 First FDEM survey and soil characterization

The initial areal FDEM survey clearly shows the presence of systematic conductive anomalies in the center of the field (Fig. 2a), i.e. four 70 m long corridors parallel to each other and approximately 8 m wide, with $EC_a$ values exceeding 30 mS m$^{-1}$. In the rest of the field, $EC_a$ values are generally lower, around 6 - 10 mS m$^{-1}$, with slightly more conductive portions (20 - 25 mS m$^{-1}$) at the northern and southern extremes, where the field borders irrigation channels. The spatial variability just described in the shallowest layer (VCP0.32) propagates along the investigation depth with a very similar pattern, with conductivity ranges gradually increasing with depth. In the bottom layer (HCP1.18), a maximum increase of approximately 15 mS m$^{-1}$ is observed, with values exceeding 40 mS m$^{-1}$ in the most conductive zones, and around 20-25 mS m$^{-1}$ elsewhere. This lateral and anomalous heterogeneity motivated the clustering process that led to the identification of the four distinct homogeneous areas where to investigate further and carry out soil sampling (Fig. 3). Specifically, parallel patterns were grouped in the center of the monitored field. Less regular patterns were identified at the Northern and Southern parts of the field. Area n. 3 is the largest one (4745 m$^2$), while n. 4 the smallest one (1876 m$^2$).





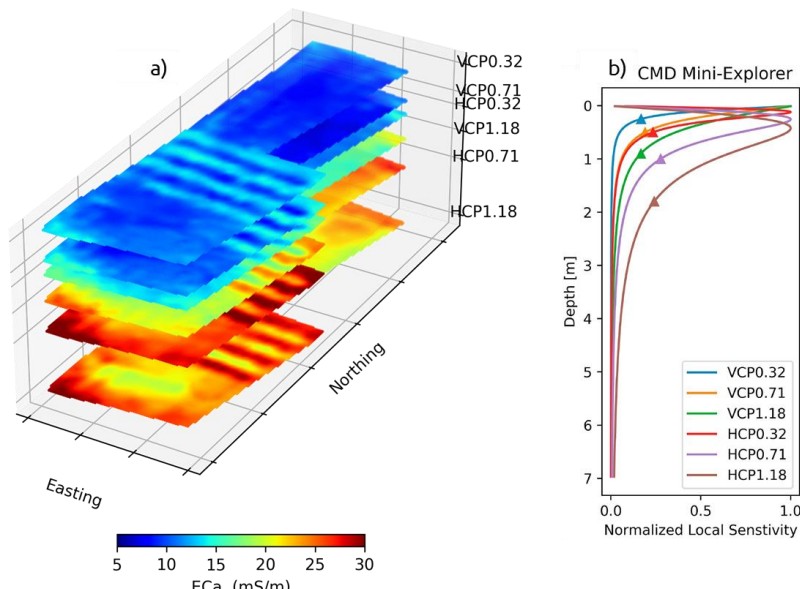


**Figure 2: a) EC$_a$ field map obtained from CMD Mini-Explorer, showing the systematic conductive seeder anomalies for each coil configuration (VCP/HCP probe orientation and Tx-Rx coil separation) and (b) their normalized local sensitivity pattern (from McLachlan et al., 2021).**

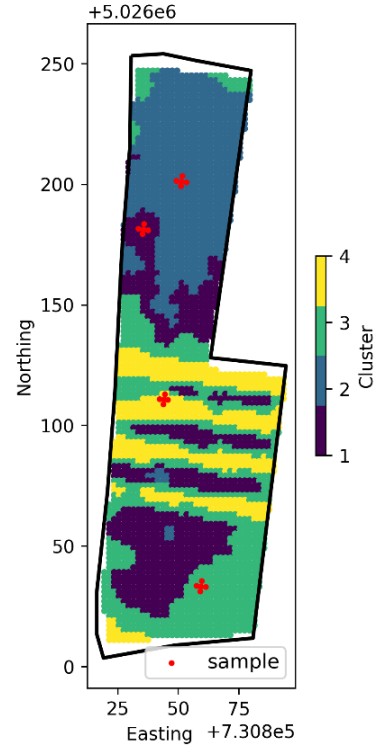

**Figure 3: a) EC$_a$ – based grouping by k-means clustering.**



Fig. 4 shows the volumetric water content map, obtained from kriging the point values measured over the entire field using the portable FDR instrument. We can clearly observe a spatial pattern very similar to $EC_a$ (Fig. 2a), characterized by the presence of systematic anomalies in the center of the field. In this case, these are portions with high water content (> 30%), arranged on parallel bands 8 m wide and the length of the field. A further area with fairly large water content (> 25%) is

located in the lower left corner, while the remaining field portions settle at values between 5 and 20%, with an average of 16.4 %.

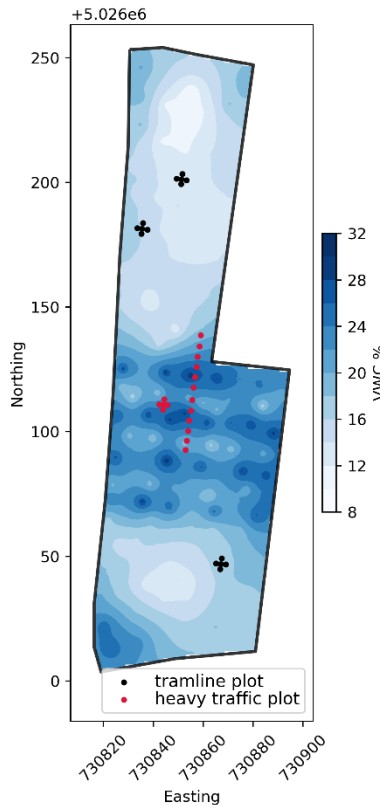

**Figure 4: VWC map obtained by Kriging of punctual TDR measurements. Samples are indicated with different colors according to the treatment.**


Seeder heavy traffic has determined significantly different soil bulk density values (significance level $p < 0.05$) compared to the rest of the field in the upper soil layers (0-0.2 m). This treatment exhibits increased BD with average values of 1.53 g cm$^{-3}$, respectively, compared to the rest of the field that averages 1.41 g cm$^{-3}$ (Fig. 5a). Along the 0.2 – 0.7 m depth profile, bulk density becomes similar between treatments, within the range of 1.56 - 1.58 g cm$^{-3}$. The volumetric water content (VWC)

shows slight differences among the treatments within the shallowest 0.2 m. The average values in this depth range are 0.26 kg kg$^{-1}$ for heavy traffic and 0.24 kg kg$^{-1}$ for uncompacted, whereas below 0.2 m depth the profiles become similar with values averaging 0.25 kg kg$^{-1}$. In terms of penetration resistance, the heavy traffic regions show significantly higher





resistances than the uncompacted throughout the depth range. Down to a depth of 0.3 m, the average penetration resistance for the non-compacted soil is 1.26 MPa while increased to an average of 4.9 MPa in compacted areas. Even at greater

depths, beyond 0.4 m, seeder traffic shows significantly higher values, with average PR values more than 2.7 MPa higher than in the uncompacted area.

From the analyses of clusters 1, 2 and 4 (Fig. 5b) a difference is shown between interrow and tramline in the shallowest 0.2 m. Both PR and BD measured on the tramlines are higher (mean values of 1.6 MPa and 1.49 g cm$^{-3}$ respectively) than those on the rows (mean values of 1.0 MPa and 1.41 g cm$^{-3}$). Deeper than the topsoil (> 0.3 m), the profiles tend to become

uniform, with monotonous growth downward. VWC profile shows higher values for the tramline than for the inter-rows in the shallowest 0.1 m (0.25 kg kg$^{-1}$ for tramline and 0.23 kg kg$^{-1}$ for interrows), while below that the behavior changes with a kind of slightly reversed trend.

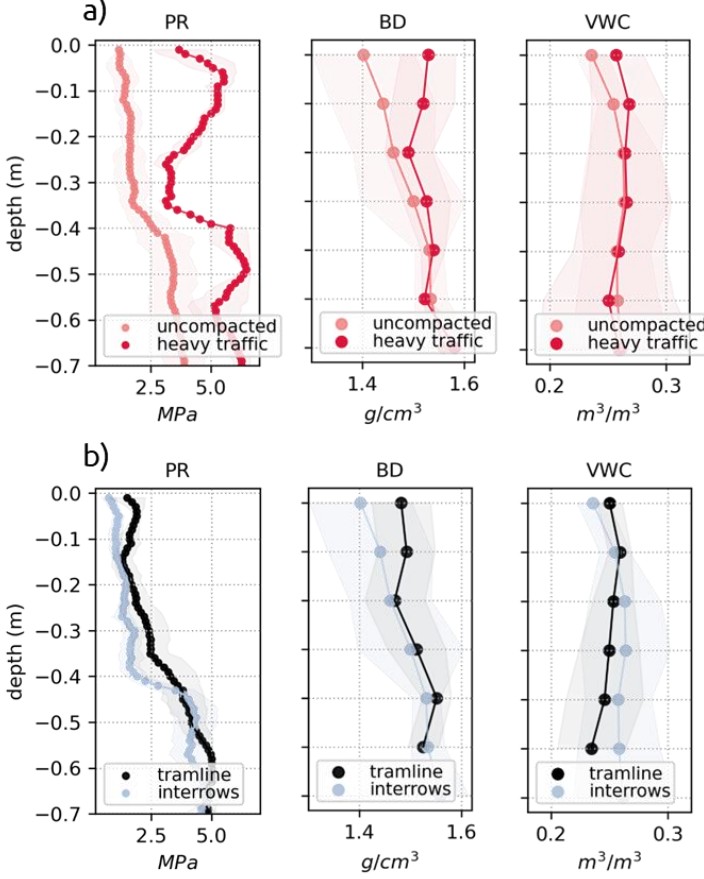

Figure 5: PR, BD and VWC reference profiles of a) heavy traffic treatment and b) tramline treatment.



## 3.2 Electromagnetic and DC-current geophysics

The three inverted transects, both FDEM and ERT, acquired across the heavy seeder traffic passages (Fig. 6), show a common pattern, consistent with the geophysical aerial survey. Two highly conductive anomalies (> 30 mS m$^{-1}$) are clearly visible in the uppermost portion of the subsoil (down to 0.5 m), located between 16 and 24 m and between 29 and 38 m along the transects direction. In the same depth range, the remaining portions of the investigated profile have an average EC value of about 9 mS m$^{-1}$. Moving to greater depths (> 0.8 m), conductivity values increase and become laterally uniform, around 40 mS m$^{-1}$. To note, the ERT transects were inverted together to generate a pseudo-3D model (Fig. 6b). In this way, the spatial extent of the systematic conductive anomalies described above could be better imaged. In addition, the 2 m electrode spacing of the ERT lines made it possible to extend the depth of investigation to approximately 6 m. However, beyond 2 m from ground level, the tomograms become uniform at values greater than 50 mS m$^{-1}$, as already found in the FDEM models (Fig. 6a).

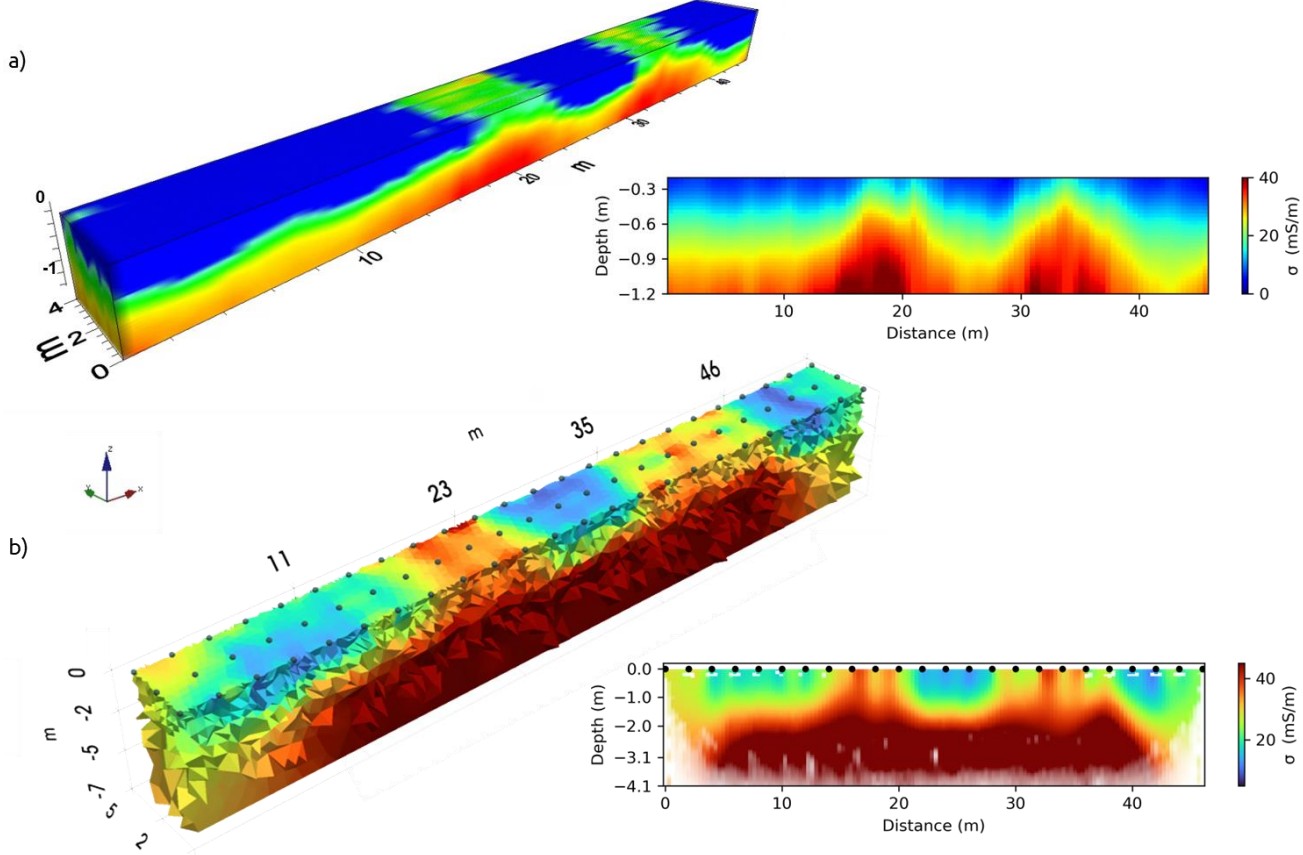

**Figure 6: Pseudo-3D inverted models obtained from the inversion of the 3 single lines, (a) FDEM and (b) ERT, over the heavy traffic area.**





As for the detailed survey, carried out in the 4 cluster areas (Fig. 3), pairs of profiles were acquired: four "lines" were superimposed on the tramlines and four "lineCs" crossed them orthogonally (Fig. 7 and 8). Figure 7 shows relatively homogeneous FDEM models, with conductivities in the range of (1 - 50 mS m$^{-1}$). Except for cluster 3, the remaining models agree in showing lower conductivity values (1 - 20 mS m$^{-1}$) in the shallow topsoil (0 – 0.5 m), gradually increasing with depth (> 30 mS m$^{-1}$) down to approximately 1.4 m. Lines 3 and 3C, acquired close to heavy traffic pathways (see Fig. 1b),

deviate from the monotonic trend just described, showing by more pronounced conductivity values (> 40 mS m$^{-1}$) at shallower depths than the previous lines. In particular, high ECs are observed as shallow as 0.3 m depth in the second portion of line3, close to the seeder passage.

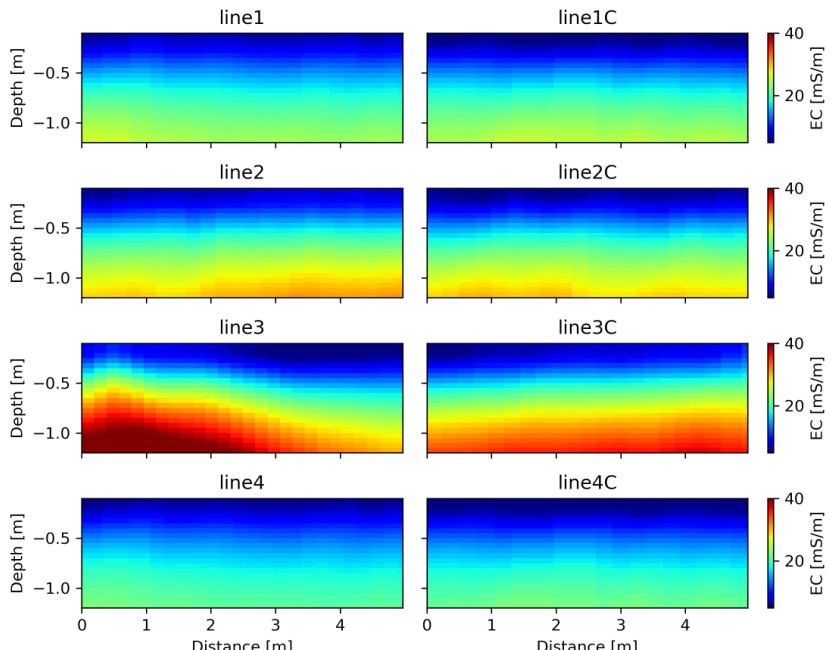

**Figure 7: EC models after FSlin inversion of FDEM trasects. The numbering is relative to the related cluster, "line" is**
**superimposed on the tramline and "lineC" crosses it orthogonally.**

ERT inverted models reflect the similarity between clusters 1, 2 and 4, and the deviation of cluster 3 (Fig. 8). As regards the former ensemble, a horizontal layer with higher conductivity (> 40 mS m$^{-1}$) with a rather homogeneous thickness is found in the uppermost portion (0 - 0.1 m). Below this, the EC values decrease (10 – 20 mS m$^{-1}$) showing basically uniform models.

The reduced spacing between the electrodes, their small number and the adopted dipole-dipole skip0 sequence contributes to increase the resolution but limits the investigation depth to only 1 m. In the transversal lines (Fig. 1), the same degree of EC homogeneity is found at depth. However, circumscribed conductive bulbs (> 40 mS m$^{-1}$) can be observed at the surface (0 - 0.1 m). They are placed at progressive distances of approximately 1 m, 3 m and 4.5 m, coinciding with the 3 tramlines intersected by these cross profiles.



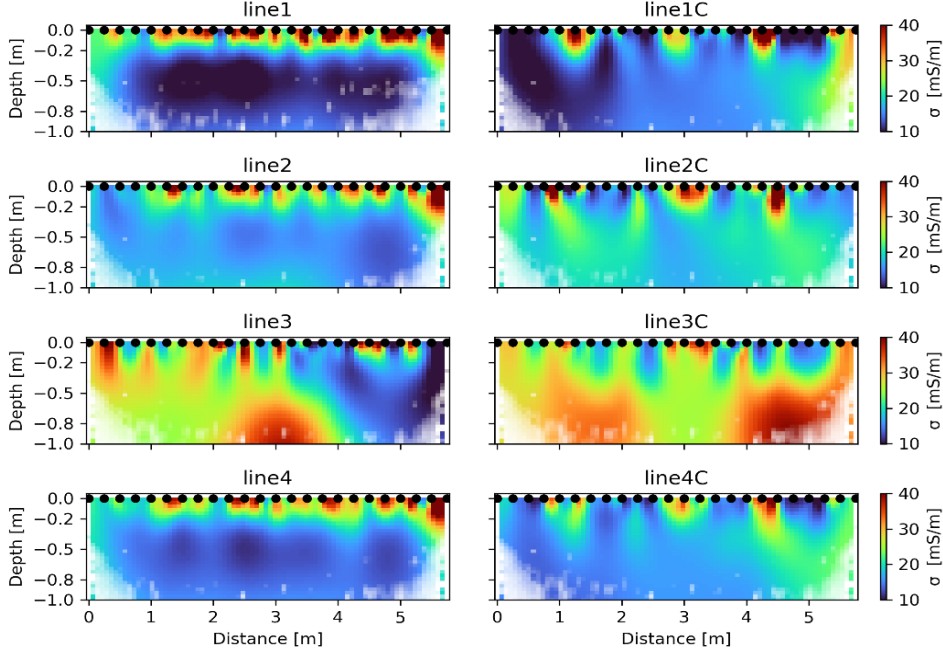


**Figure 8: ERT inverted models of detailed transects. The numbering is relative to the related cluster, "line" is superimposed on the tramline and "lineC" crosses it orthogonally.**

### 3.3 Primary soil properties and electromagnetic behavior

When analyzing the relationships between primary soil parameters, a significant correlation was found between PR and BD ($r_s = 0.23$), and VWC and BD ($r_s = 0.43$, $p < 0.01$), while no dependency was found between the texture and other physical properties (Fig. 9a). VWC was also correlated with electrical conductivity in topsoil, both with $EC_{FDEM}$ ($r_s = 0.28$) and $EC_{ERT}$ ($r_s = 0.32$) Fig. 9a). Similarly, conductivity demonstrated high significant correlations with PR ($r_s = 0.41$ and $0.39$ for $EC_{FDEM}$ and $EC_{ERT}$, respectively) whereas only $EC_{ERT}$ exhibits a positive correlation with BD ($r_s = 0.32$), differing from

$EC_{FDEM}$ in this case.

An EC dependency from VWC was also found when the moisture content of the field topsoil (down to 0.2 m) was considered. The VWC measured with portable TDR in 128 randomly distributed samples (Fig. 1c), strongly influenced the areal electromagnetic response ($r_s = 0.78$), as reported in Fig. 9b.






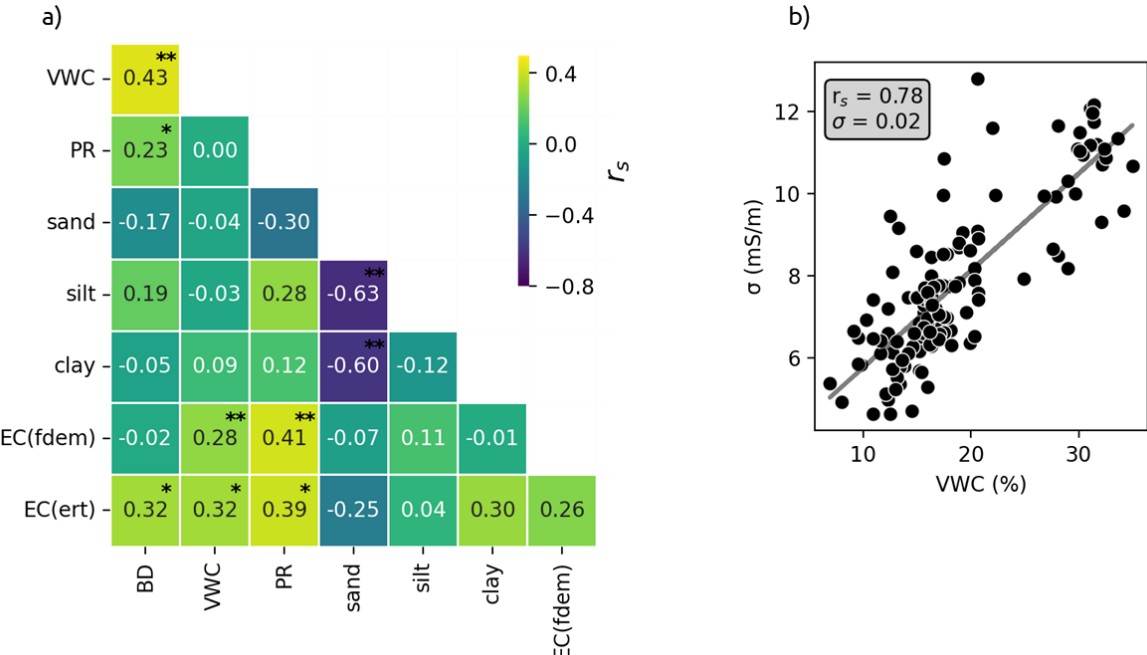

**Figure 9: a) Heatmap displaying the Spearman correlation coefficients ($r_s$) between the measured variables (BD = bulk density, VWC = volumetric water content, PR = penetration resistance, sand, silt, clay, EC(fdem) = electrical conductivity from FDEM and EC(ert) = electrical conductivity from ERT). The asterisk identifies significant relationships, * at p < 0.05 and ** at p < 0.01. b) Spearman's correlation between volumetric moisture content (measured with TDR) and $EC_{FDEM}$ from the areal survey.**

## 4 Discussion

This work explores the capabilities of ERT and FDEM to discriminate soil compaction in the field, both at an aerially extended and detailed scales, and provides some methodological insights to optimize geophysical acquisition for this specific goal. Since EC is a parameter influenced by multiple soil properties (Friedman, 2005; Doolittle and Brevik, 2014), it is useful to explore its correlation with soil bulk density (BD), penetration resistance (PR) and volumetric water content (VWC) in order to improve soil management and monitoring of agricultural practices.

As a starting point, we utilized BD and PR as indicators of the state of compaction. Soil BD is the mass of dry soil per unit volume obtained from a sample, while PR is measured in the field as the result of the cohesive forces between the individual soil particles and the frictional resistance encountered by the particles sliding over each other (Marshall and Holmes, 1980). Although often affected by punctual heterogeneity when measured in the field, PR remains an effective indicator of soil compaction (Benevenute et al., 2020). A poorly-structured soil is characterized by high BD values (Håkansson and Lipiec, 2000), and this often happens with the repeated passage of heavy vehicles in unfavorable field conditions. As in our case, the portions of soil heavily compacted on the surface for heavy seeder passes show very high BD (> 1.55 g cm⁻³) and PR (> 4.5 MPa). Even when considering the case of common tramline compaction, the values are higher than uncompacted by roughly





1.5 g cm$^{-3}$ and 2.2 MPa, respectively, in line with findings of Reintam et al. (2009) and Elaoud and Chehaibi (2011) for this type of soils.

Taking VWC into consideration as well, an increase in moisture is observed in the compacted subsurface portions. The behavior is definitely evident in the areal mapping performed with TDR (Fig. 4), but it is also observed from the samples collected from 0 - 0.3 m depth layer ($r_s$ = 0.43 with BD, Fig. 9b). This phenomenon generates where the percolation rate of

rainwater or irrigation water through the soil is reduced by a compacted layer, thus developing localized saturated zones close to the soil surface (Batey and McKenzie, 2006).

This aspect is of paramount importance when evaluating the geophysical response, particularly considering the electromagnetic nature of the methods here (and commonly) adopted. Indeed, they rely on electro-magnetic properties of the subsurface, which change dramatically with water content (Binley et al., 2015; Vereecken et al., 2007). In fact, a positive

correlation was observed between electrical conductivity and soil moisture in the areal survey ($r_s$ = 0.78 for EC$_{FDEM}$, Fig. 9a), but also a significant relationship by examining soil samples ($r_s$ = 0.28 and 0.32 for EC$_{FDEM}$ and EC$_{ERT}$ respectively, Fig. 9b). In both areal and detailed surveys, the highly compacted portions of the soil are characterized by high electrical conductivity anomalies relative to the context. This aspect may therefore be indicative of soil compaction if electro-magnetic surveys are used as an initial monitoring tool. In general, the conductivities found are in the range below 50 mS m$^{-1}$, in good agreement

with the agricultural context of the area. FDEM models agree in showing lower values (below 10 mS m$^{-1}$) in the shallowest soil layer (0-0.1 m), likely due to drier conditions, with a gradual increase with depth (>30 mS m$^{-1}$), related to an increase in water content (Fig. 7). This dynamic is also evident in the pseudo 3D models in Fig. 6, for both ERT and FDEM results. Note that the circumscribed and systematic surface conductive anomalies present here (> 40 mS m$^{-1}$), are precisely placed on the seeding corridors that compacted the first subsurface.

The high-resolution ERT transects are informative down to depths of about one meter (Fig. 8), as they are designed to achieve a high resolution close to the surface, where they show localized conductive anomalies (about 0.3 m wide, 0.15 m thick) in correspondence of the compaction generated on the tramlines. This resolving ability is not found in FDEM transects, and this is motivated by the very nature of the techniques: the electromagnetic induction approach generates vertical soundings with integrated EC$_a$ values over depth (McNeill, 1980) and the output models are strictly dependent on the

instrument footprint, the spatial density of the measurements, and the need to smooth the measured data during processing and inversion. Given the adopted instrument, it is expected that its footprint be not smaller than a square meter. On the contrary, ERT can be adapted to the scale and resolution needed to optimally investigate the phenomenon of interest with a tomographic approach. The different surface sensitivity of the two methods to compaction is also confirmed by statistical analysis: EC$_{ERT}$ shows a significant correlation with BD ($r_s$=0.32) whereas EC$_{FDEM}$ is even slightly negative ($r_s$=-0.02) One

last thing to consider, in this specific field of study concerned with the most proximal portion of the subsurface (down to 1-2 m), is the critical effect of a correct placing of the electrodes: they should penetrate not too far into the soil to ensure that the assumption of punctual current injection is satisfied, given the commonly small spacings; however, at the same time, good coupling with the plowed and aerated soil must be ensured, minimizing contact resistances.



The advantages of FDEM instruments are considerable, ease of use and speed of acquisition first and foremost. The
equipment is commonly lightweight (a CMD Mini-Explorer probe weighs 2 kg), and measurements are collected contactless
by simply carrying the device over the target area, walking or dragging it. With little effort and in less than 2 hours, the $EC_a$
distribution in the shallow subsurface can be acquired over about 2 hectares, thus highlighting portions with high compaction
and surface moisture. However, as shown here but also noted by Blanchy et al. (2024), detailed spatial heterogeneities at
both areal and depth scales can be missed by FDEM instruments. Other important aspects are (a) during acquisition to avoid
the nearby presence of metal objects that can produce spurious results, and (b) check the potential instrumental drift (De
Smedt et al., 2016). In addition, it is crucial to remember that in the presence of a conductive soil, most of the signal at
higher frequencies is conveyed, via electromagnetic induction, in the topmost layer, decreasing the depth of investigation.
Therefore, although ERT requires more logistical effort and the need for galvanic contact with the soil, it remains a
technique of fundamental application for obtaining a more accurate subsurface model with sufficient spatial resolution. In
this respect, the existence of new georesistivimeters (i.e. Syscal Terra, Iris Instruments) capable of collecting datasets with
moving streamer systems could generate a breakthrough to obtain truly 3D field electrical conductivity models.

Our results show a good correlation between FDEM and ERT in highlighting the compacted and saturated portions of the
soil, with some differences. Despite its potential and widespread application, the use of electromagnetic geophysics in
agriculture presents challenges. Survey resolution, adaptability, and ease of use are all paramount. Also, soil electrical
conductivity depends on soil type, moisture content and texture, highlighting the complexity inherent in understanding which
factor is predominant in a site-specific situation. Identifying compaction through EC measurements, therefore, is not a trivial
task. Variability in field conditions, sensor calibration and data interpretation are critical considerations. This requires multi-
parameter approaches that incorporate direct point measurements (e.g., bulk density, porosity and permeability, soil
penetration resistance) (Johnson and Bailey, 2002; Keller et al., 2021) and auxiliary data like historical land use. However,
the spatial extent and resolution permitted by non-invasive geophysical methods is a great advantage over other local
measurement methods, and increased efforts shall be devoted towards improving accuracy of such techniques in identifying
soil compaction together with other parameters important in soil management.

## 5 Conclusions

In this work, we compare the efficiency of the two geophysical techniques most used in agronomy, i.e. ERT and FDEM, and
specifically for the characterization of soil compaction. We specifically explore the sensitivity and resolution of these
methods in assessing shallow soil compaction in the field, at both plot and submeter-tramline scale.

FDEM allows rapid acquisition of measurements that can define spatial variability at the ground surface, which motivates its
appreciation in soil science and agronomy. Nevertheless, it must be noted that a rigorous acquisition protocol must be
applied in order to avoid potential instrumental drift and other issues e.g. related to local strong conductors, and, therefore,
scattered or negative values and local conductivity anomalies unrelated to soil structure. Moreover, FDEM inversion is for



now a purely one-dimensional process with depth, with lower spatial resolution as compared to ERT. In addition, FDEM has an intrinsic spatial scale linked to the coil distance and used frequency, that can hardly achieve a resolution finer than the meter scale.

Due to its accuracy and flexibility of application, the ERT method is well established and widespread as well in agronomy.
However, it also requires a rigorous approach to ensure the desired resolution and reliability of result. The type of measurement sequence, site- and target-specific, is known to be critical, and reciprocal data acquisition is strongly recommended since it allows an assessment of the quality of the collected dataset, and therefore provides tools to minimize possible artifacts in the reconstructed subsurface models.

Future challenges must address the increase in spatial resolution and sampling potential of electromagnetic induction
instruments, as well as the development of algorithms that could allow a true 3D inversion of the spatially measured data. At the same time, the use of next generation georesistivimeters capable of collecting datasets with moving streamer systems represents a great opportunity to be tested in the field. Technology advances are visible in this direction, and this will foster precision agriculture practices but also a broader understanding of soil-plant-water interactions and ecosystem dynamics.

**Data availability**

Data used to obtain the results presented in this work can be accessed on ZENODO open-source repository (DOI: 10.5281/zenodo.11356672).

**Author Contribution**

Conceptualization: AC. Methodology: AC, GC, FM. Validation: GC. Investigation: AC. Data curation: AC, ML, LP. Writing - original draft preparation: AC. Writing - review and editing: AC, ML, LP, GC, FM. Funding acquisition: FM.

**Competing interests**

The authors declare that the research was conducted in the absence of any commercial or financial relationships that could be
construed as a potential conflict of interest.

**Acknowledgments**

Authors thank Franco Gasparini, Riccardo Polese and Ilaria Piccoli for their valuable support on fieldwork assistance. The research that led to these results received funding from both the European Union's Horizon 2020 and the MIPAAF - Ministry
of Agricultural Food and Forestry Policies research and innovation program (grant agreement no. 862665 ICT-AGRI-FOOD, SoCoRisk project), and the Agritech National Research Center (Spoke 4, WP4.3), European Union Next-Generation EU (PIANO NAZIONALE DI RIPRESA E RESILIENZA (PNRR) – MISSIONE 4 COMPONENTE 2, INVESTIMENTO 1.4 – D.D. 1032 17/06/2022, CN00000022). This manuscript reflects only the authors' views and opinions, neither the European Union nor the European Commission can be considered responsible for them.



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
