# Peer review of "Uncovering soil compaction: performance of electrical and electromagnetic geophysical methods"

_EGUsphere, 2024_

## Referee Comment (RC1)

Rapid methods for assessment of soil compaction at various spatial scales are much needed and near-surface geophysics is increasingly becoming popular to address this challenge. In this work, the authors compare and contrast the suitabilities and limitations of electromagnetic induction and electrical resistivity tomography methods for assessing soil compaction by considering the spatial resolution and scale aspects. I commend the authors for this valuable contribution towards managing expectations on sensors' efficacy and I believe this article is an excellent fit for the special issue on Agrogeophysics. I suggest the authors do a moderate revision before it can be accepted for publication.

General comments:

1) Please refrain from using abbreviations in the title, figure captions and at the beginning of the sentences. Please define abbreviations before their first usage. Also, please use the same terminology for the sensing technology consistently, e.g., replace DC-current with electrical resistance tomography (ERT).
2) The abstract needs to be more focused and highlight the work's unique contribution. Please consider revising.
3) Kindly improve the figures especially Fig. 1 and 2. In Fig. 1, it would be nice to see the location of the inset in Fig. b., also please change "FDR" to "TDR" in the legend. Are the tramlines orthogonal to the seeder traffic? In Fig. 2a, please present the maps in 2D rather than 3D. In the current form, it is difficult to see the ECa variability in deeper measuring channels.

Specific comments:

1) In lines 45-50, "moisture EC-derived content" should be "EC-derived moisture content".
2) In lines 115-120, "Electro-Magnetic Induction" should be "Electromagnetic induction".
3) In lines 140-145, "Cumulative Sensity (CS)" should be "Cumulative Sensitivity (CS)". I think it would also be nice to include a reference here on the inversion codes based on the CS forward model.
4) In lines 145-150, "8 cross-transects". It would be nice to see them in the Fig. 1.
5) In line 170, "discharging" should be "removing"?
6) In lines 185-190, "The number of homogeneous areas were automatically selected resulting in four clusters". This statement needs further explanation on what basis. Is it the Elbow method or the Silhouette score?
7) In lines 205-210, "Indeed, EC soil properties showed a non-normal distribution." This sentence needs to be revised. Do you mean you used Spearman's coefficient because you expect the relationship to be monotonous?
8) In lines 230-235, "FDR" should be "TDR".
9) In lines 300-305, Please see if you find any rule-of-thumb to define categories from strong to weak.
10) In lines 370-375, "In addition, it is crucial to remember that in the presence of a conductive soil, most of the signal at higher frequencies is conveyed, via electromagnetic induction, in the topmost layer, decreasing the depth of investigation." Do you mean that most of the currents stay within the top soil and do not diffuse to the subsoil?

All the best!

---

## Referee Comment (RC3)

General summary

The manuscript presents a comprehensive comparison between two geophysical techniques, Electrical Resistivity Tomography (ERT) and Frequency-domain electromagnetic Method (FDEM), for assessing soil compaction in agricultural contexts. It effectively highlights the strengths and limitations of both methods, with a focus on their resolution and efficiency in characterizing soil structure at various scales. The study offers valuable insights into the applicability of these non-invasive methods for improving soil monitoring in agriculture, contributing to the optimization of geophysical data acquisition and processing. The results are well-supported by traditional soil characterization techniques and validated with traditional soil characterization techniques (penetration resistance, bulk density, volumetric water content), making the findings relevant for advancing agriculture practices and enhancing the understanding of soil-plant-water interactions.

I would like to acknowledge the authors for their contribution to addressing the practical limitations and potential of sensor technologies in soil monitoring. This manuscript is a strong candidate for inclusion in the special issue: Agrogeophysics, as it provides important insights into the performance of geophysical methods in agricultural applications. I recommend a minor revision to further refine the clarity and presentation of the findings.

I appreciate the authors for their valuable contribution to setting realistic expectations regarding sensor efficacy. I believe this manuscript is suitable for inclusion in EGUsphere, as it provides important insights into the performance of geophysical methods in agricultural applications. However, I believe that there are some points that can be improved. Therefore, I recommend a major revision.

All the best!

General comments –

1. Please revise the abstract to more clearly reflect the specific objectives, methods and findings of the study.
2. Please review and correct the usage of abbreviations throughout the manuscript.
3. Please re-check the figures and figure captions.

Specific comments –

Title – Suggest modifying the title as ""Electrical and electromagnetic geophysics for soil compaction assessment"

Abstract – Lines 5 – 15: the introduction sentences of the abstract should be specific to the present study

Lines 10 – 15 – "agricultural soil", Please specify the soil – "silt loam"

Lines 10-15 – Please refine the objective in the abstract specific to the study

Lines 15 – 20 – Please provide an overall methodology statement (one sentence) including the methods of different analysis (correlations, K means clustering etc.)

Lines 20 – 25 – Please include the overall finding of this study in the abstract.

Lines 30 – 35 – "Soil properties, agricultural processes, and moisture dynamics." Please include state variables as well to represent soil water content and soil salinity.

Lines 45 – 50 – "EC" please define the abbreviation in the first place

Lines 60 – 65 – "Soil electrical conductivity (EC)" should be soil EC

Lines 60 – 65 – Soil EC is also used to estimate soil water content and soil salinity – please include that as well

Lines 75 – 80 – "with increased electrical conductivity" should be "with increased EC"

Lines 80 -85 – "for the assessment of soil surface compaction." – readers would like to know the depth, if possible, please provide the depth range within brackets.

Lines 80 85 – "The survey was conducted both at the field scale, covering an area of 1.5 hectares, and in detail on individual targeted transects". Please rephrase the sentence for clarity.

Lines 90 -95 – "Results, validated with direct information, show the pros & cons of both FDEM and ERT techniques and how differences in their spatial resolution heavily influence their ability to characterize compacted areas with good confidence." This content does not fit here. Please remove or modify.

Figure 1 – Fig. 1b – "FDR"? Is this "TDR"?

Fig. 1c - It is difficult to understand where this matches with Figure 1. B, please modify the figures and clearly show the figure c in figure b.

Heading 2.2 – Please remove the abbreviation – already introduced

Lines 115 – 120 – "Electro-Magnetic" should be electromagnetic

Lines 120 – 125 – "electrical conductivity (EC)" already introduced – please use EC

Lines 120 – 125 – "can be probed" please replace as "can be obtained"

Line 130 – "4m" should be "4 m"; please keep a space between the number and the unit.

Heading – 2.3 – Please remove the abbreviation

Lines 160 -165 – "first few centimetres of the soil", please provide a number, how many centimetres.

Lines 165 – 170 – "Q=2%" Please provide what is "Q"?

Lines 175 – 180 – "Both geo-electric and electromagnetic," Please use a consistency term to represent ERT. For example, "Both electrical and electromagnetic"

Lines 175 – 180 – In the Fig. 1C it is difficult to understand the discussed lines in this sentence "an initial areal FDEM acquisition was followed by 3 additional lines to intercept seeder heavy passages, and 8 detailed transects, both FDEM and ERT (4 along and 4 across normal tractor tramlines. Please modify the Fig. 1C for clarity.

Line 185 – "*dij" should be "dih"?*

Lines 185 – 190 – "For each area, one geophysical detailed survey (i.e. ERT + FDEM) was performed." Please rewrite this sentence for clarity. One geophysical detailed survey or both surveys (ERT + FDEM)?

Lines 190 – 195 – "throughout the 0–80 cm soil layer." Please provide the soil sampling depth intervals.  Same in " 0.70 m" as well.

Lines 195 – 200 – "bulk density" should be "BD", "4m" should be "4 m"

Lines 200 – 205 – "22cm" should be "22 cm"

Heading 3.1 – What is mean by "First" here

Lines 215 – 220 – Please introduce the abbreviations (VCP0.32 and HCP1.18) in the first use.

Lines 220 – 225 and Figure 2a - I'm just wondering why the legend of Fig 2a ranges only to 30 mS/m and not 40 mS/m? In the content authors mentioned "In the bottom layer (HCP1.18), a maximum increase of approximately 15 mS $m^{-1}$ is observed, with values exceeding 40 mS $m^{-1}$ in the most conductive zones,"

Figure 2.a – Please rearrange the overlapped labels

Figure 3 – "a" should be removed. The sample symbol in the legend is different from the map.

Lines 230 – 235 – "FDR" should be "TDR"

Figure 4 – VWC please introduce the abbreviation first.

Lines 240 – 245, and 255 – 260 – Why the unit for water content in kg/kg Please correct the unit of volumetric water content to $m^3/m^3$, as similar to the figure 5.

Lines 320 -325 – Please remove the already introduced abbreviations.

Lines 340 -345 – "In both areal and detailed surveys, the highly compacted portions of the soil are characterized by high electrical conductivity anomalies relative to the context." What would be the reason for this observation, please discuss.

Lines 375 – 380 – That would be nice if the authors could mention the most important challenges here.

"Despite its potential and widespread application, the use of electromagnetic geophysics in agriculture presents challenges such as......"

---

## Author Response (AR1)

**EGUSPHERE-2024-1587 | Replies to Reviewers**

Dear Editor,

we hereby submit a revised version of our manuscript EGUSPHERE-2024-1587. We sincerely thank you and the reviewers for your precious contributions to improve our manuscript and all the positive feedbacks received.

While developing this version, we revised the initial manuscript following Editor's and Reviewer's comments, and consequently, incorporated all the improvements suggested. In particular:

i)      we clarified the objectives of the manuscript -namely, to compare and contrast the suitability and limitations of ERT and FDEM techniques for soil compaction assessment, considering the spatial resolution and scale aspects- improving the abstract and modifying the title itself,

ii)     we corrected the whole manuscript to be consistent in the terminology and in the use of abbreviations,

iii)    we improved the figures to make them clearer and more understandable, replacing, where present, the "jet" colormap with the "turbo", fine-tuned to remedy false details, banding, and color-blind ambiguity, but at the same time maintaining the accentuation of details which allows faster visual assessment https://research.google/blog/turbo-an-improved-rainbow-colormap-for-visualization/ .

In the following pages, we present our response to each of the comments raised in the review process. Please consider that the reference lines refer to the clean version of the revised manuscript. No doubt, your comments have helped to improve our manuscript, hopefully to a degree suitable for publication in *SOIL.*

**Editor**

Three reviewers have now reviewed this manuscript, all considered this work valuable and suitable for publication in this special issue. The reviewers also provided several constructive comments to improve the quality of the manuscript.

I agree with the generally positive assessment of the manuscript and its suitability for SOIL. I consider the authors' response to be appropriate and that authors can successfully address the reviewers' concerns. Therefore, I recommend considering this manuscript after these revisions. I suggest authors to particularly address the issue of clarifying the objectives of the manuscript to help managing expectations of the readers. As currently the title and abstract suggest that the manuscript provides a wider/systematic overview of DC resistivity and EMI for soil compaction detection, but the study focuses on an experimental site with rather controlled conditions. In addition, I recommend to consider changing the color maps from rainbow to a perceptually uniform color map (see https://doi.org/10.1038/s41467-020-19160-7).

All the best,

Alejandro

**Referee #1**

*Rapid methods for assessment of soil compaction at various spatial scales are much needed and near-surface geophysics is increasingly becoming popular to address this challenge. In this work, the authors compare and contrast the suitabilities and limitations of electromagnetic induction and electrical resistivity tomography methods for assessing soil compaction by considering the spatial resolution and scale aspects. I commend the authors for this valuable contribution towards managing expectations on sensors' efficacy and I believe this article is an excellent fit for the special issue on Agrogeophysics. I suggest the authors do a moderate revision before it can be accepted for publication.*

*All the best!*

General comments:

1.  Please refrain from using abbreviations in the title, figure captions and at the beginning of the sentences. Please define abbreviations before their first usage. Also, please use the same terminology for the sensing technology consistently, e.g., replace DC-current with electrical resistance tomography (ERT).
    We checked and corrected the whole manuscript to be consistent in the terminology of the techniques and in the use of abbreviations.

2.  The abstract needs to be more focused and highlight the work's unique contribution. Please consider revising.
    Thanks for the comment, we rephrased the abstract highlighting the innovative aspects of the work (LL 14-27)

3.  Kindly improve the figures especially Fig. 1 and 2. In Fig. 1, it would be nice to see the location of the insert in Fig. b., also please change "FDR" to "TDR" in the legend. Are the tramlines orthogonal to the seeder traffic? In Fig. 2a, please present the maps in 2D rather than 3D. In the current form, it is difficult to see the ECa variability in deeper measuring channels.
    Thanks for the suggestions. We improved Fig. 1 and its caption (LL 100-103). The tramlines are longitudinal to the field, while seeder traffic occurred transversely, but not exactly orthogonal since they are separate experiments.
    As for Fig. 2, we replaced the 3D visualization with 2D ECa maps referring to each coil.

Specific comments:

| Comment | Reply |
|---|---|
| In lines 45-50, "moisture EC-derived content" should be "EC-derived moisture content". | Replaced, thank you (L 51). |
| In lines 115-120, "Electro-Magnetic Induction" should be "Electromagnetic induction". | Replaced, thanks (L 118). |
| In lines 140-145, "Cumulative Sensity (CS)" should be "Cumulative Sensitivity (CS)". I think it would also be nice to include a reference here on the inversion codes based on the CS forward model. | Replaced and reference added, thank you (L 145). |
| In lines 145-150, "8 cross-transects". It would be nice to see them in the Fig. 1. | Actually, the 8 cross-transects are already shown in Fig. 1B at numbers 1 to 4. Being short lines, it is difficult to appreciate |

| | them sharply without covering the samples position in the field plot. Therefore, a zoom frame of their arrangement is shown in Fig. 1C |
|---|---|
| In line 170, "discharging" should be "removing"? | Corrected, thank you (L 171). |
| In lines 185-190, "The number of homogeneous areas were automatically selected resulting in four clusters". This statement needs further explanation on what basis. Is it the Elbow method or the Silhouette score? | The number of groups was determined using the Calinski-Harabasz index, which quantifies the ratio of between-group variance to within-group variance, and ensures an optimal balance between group distinctiveness and internal cohesion. Details have been added to the text (L189-191). |
| In lines 205-210, "Indeed, EC soil properties showed a non-normal distribution." This sentence needs to be revised. Do you mean you used Spearman's coefficient because you expect the relationship to be monotonous? | Revised and rephrased, thanks (LL 209-210) |
| In lines 230-235, "FDR" should be "TDR". | Corrected, thanks (L 234). |
| In lines 300-305, Please see if you find any rule-of-thumb to define categories from strong to weak. | We modified the text according to the present correlations categories used as rule-of-thumb:
▪ 0 - ±0.3     weak
▪ ±0.3 - ±0.7   moderate
▪ ±0.7 - ±1     strong
(LL 303-311) |
| In lines 370-375, "In addition, it is crucial to remember that in the presence of a conductive soil, most of the signal at higher frequencies is conveyed, via electromagnetic induction, in the topmost layer, decreasing the depth of investigation." Do you mean that most of the currents stay within the topsoil and do not diffuse to the subsoil? | Exactly, we rephrased to make it clearer (LL 374-375). |

**Referee #2**

*The use of geophysical methods in the field of agriculture is gaining in popularity, as this article demonstrates. The authors compare two geophysical methods: Electrical Resistivity Tomography (ERT) and the Frequency Domain Electromagnetic Method (FDEM), to assess the state of the soil as a result of agricultural practices that increase its compaction and compromise its hydromechanical properties. Knowing the state of agricultural soil is of paramount importance to better understand how to manage future cultivation, especially in terms of irrigation and nutrients. I thank the authors for this important work that demonstrates how geophysical techniques can be of great support in evaluating agricultural practices and especially in knowing the state of the soil, in order also to reduce water wastage without compromising the physical state of the soil.*

*The article is well structured, and I find it suitable for the special issue on Agrogeophysics.*

*Congratulations on a job well done and all the best!*

Some suggestions:

| Comment | Reply |
|---|---|
| Emphasize in the abstract the importance and innovative aspect of the work. | Thanks for the comment, we rephrased the abstract highlighting the innovative aspects of the work (LL 15-27) |
| Improve Fig1, showing in the legend the meaning of everything in the figures. | We improved Fig. 1 adding all the elements in the legend and further information in the description (LL 100-103) |
| Indicate in the figure the transects on which the FDEM surveys were carried out. | Fig. 1 already shows the location of all FDEM transects acquired with bright purple color, which is exactly the same as ERT transects. We have already made this explicit in LL 177-179, and included more details in the description of Fig. 1 as suggested. |
| Line 230-235 Sure FDR? The volumetric content was measured with a TDR sensor. | Corrected, thanks (L 234). |

**Referee #3**

*The manuscript presents a comprehensive comparison between two geophysical techniques, Electrical Resistivity Tomography (ERT) and Frequency-domain electromagnetic Method (FDEM), for assessing soil compaction in agricultural contexts. It effectively highlights the strengths and limitations of both methods, with a focus on their resolution and efficiency in characterizing soil structure at various scales. The study offers valuable insights into the applicability of these non-invasive methods for improving soil monitoring in agriculture, contributing to the optimization of geophysical data acquisition and processing. The results are well-supported by traditional soil characterization techniques and validated with traditional soil characterization techniques (penetration resistance, bulk density, volumetric water content), making the findings relevant for advancing agriculture practices and enhancing the understanding of soil-plant-water interactions.*

*I would like to acknowledge the authors for their contribution to addressing the practical limitations and potential of sensor technologies in soil monitoring. This manuscript is a strong candidate for inclusion in the special issue: Agrogeophysics, as it provides important insights into the performance of geophysical methods in agricultural applications. I recommend a minor revision to further refine the clarity and presentation of the findings. I appreciate the authors for their valuable contribution to setting realistic expectations regarding sensor efficacy. I believe this manuscript is suitable for inclusion in EGUsphere, as it provides important insights into the performance of geophysical methods in agricultural applications. However, I believe that there are some points that can be improved. Therefore, I recommend a major revision.*

*All the best!*

General comments:

1. Please revise the abstract to more clearly reflect the specific objectives, methods and findings of the study.
   Thanks for the comment, we rephrased the abstract highlighting the innovative aspects of the work (LL 15-27)

2. Please review and correct the usage of abbreviations throughout the manuscript
   We checked and corrected the whole manuscript to be consistent in the use of abbreviations.

3. Please re-check the figures and figure captions.
   Thanks for the suggestions. We modified some figures (Fig.1, Fig.2) as already suggested by Reviewer #1 and improved their captions.

Specific comments:

| Comment | Reply |
|---|---|
| Title – Suggest modifying the title as ""Electrical and electromagnetic geophysics for soil compaction assessment" | Thanks for the suggestion, we improved the title to be consistent with the aims and the terminology. |
| Abstract – Lines 5 – 15: the introduction sentences of the abstract should be specific to the present study | We believe that we have briefly introduced the subject of this study (i.e., soil structure monitoring and specifically soil compaction), in a few lines, and placed it in a broader context before |

| | |
|---|---|
| | describing the highlights of the work. We therefore consider keeping this structure. |
| Lines 10 – 15 – "agricultural soil", Please specify the soil – "silt loam" | Corrected, thanks (L 16). |
| Lines 10-15 – Please refine the objective in the abstract specific to the study | Thanks for the comment, we rephrased the abstract highlighting the innovative aspects of the work (LL 15-27) |
| Lines 15 – 20 – Please provide an overall methodology statement (one sentence) including the methods of different analysis (correlations, K means clustering etc.) | Added, thanks (LL 17-18). |
| Lines 20 – 25 – Please include the overall finding of this study in the abstract. | Added, thanks (LL 19-27). |
| Lines 30 – 35 – "Soil properties, agricultural processes, and moisture dynamics." Please include state variables as well to represent soil water content and soil salinity. | Added, thanks (L 33). |
| Lines 45 – 50 – "EC" please define the abbreviation in the first place | Corrected, thanks (L 48). |
| Lines 60 – 65 – "Soil electrical conductivity (EC)" should be soil EC | Corrected, thanks (L 62). |
| Lines 60 – 65 – Soil EC is also used to estimate soil water content and soil salinity – please include that as well | Added, thanks (LL 62-63). |
| Lines 75 – 80 – "with increased electrical conductivity" should be "with increased EC" | Corrected, thanks (L 80). |
| Lines 80 -85 – "for the assessment of soil surface compaction." – readers would like to know the depth, if possible, please provide the depth range within brackets. | Added, thanks (L 84). |
| Lines 80 85 – "The survey was conducted both at the field scale, covering an area of 1.5 hectares, and in detail on individual targeted transects". Please rephrase the sentence for clarity. | The sentence has been rephrased, thanks (LL 84-85). |
| Lines 90 -95 – "Results, validated with direct information, show the pros & cons of both FDEM and ERT techniques and how differences in their spatial resolution heavily influence their ability to characterize compacted areas with good confidence." This content does not fit here. Please remove or modify. | The sentence has been modified (LL 91-92). |
| Figure 1 – Fig. 1b – "FDR"? Is this "TDR"? | Exactly, corrected (Fig. 1b). |

| | |
|---|---|
| Fig. 1c - It is difficult to understand where this matches with Figure 1. B, please modify the figures and clearly show the figure c in figure b. | We improved Fig. 1 and added further information in the description (LL 100-103) |
| Heading 2.2 – Please remove the abbreviation – already introduced | Removed, thanks (L 115). |
| Lines 115 – 120 – "Electro-Magnetic" should be electromagnetic | Corrected, thanks (LL 115-120). |
| Lines 120 – 125 – "electrical conductivity (EC)" already introduced – please use EC | Corrected, thanks (L 120). |
| Lines 120 – 125 – "can be probed" please replace as "can be obtained" | Corrected, thanks (L 124). |
| Line 130 – "4m" should be "4 m"; please keep a space between the number and the unit. | Corrected consistently in the whole manuscript, thanks. |
| Heading – 2.3 – Please remove the abbreviation | Removed (L 150). |
| Lines 160 -165 – "first few centimetres of the soil", please provide a number, how many centimetres. | Information provided (L 165). |
| Lines 165 – 170 – "Q=2%" Please provide what is "Q"? | Explanation was given a few words earlier (LL 169-170), it is basically the minimum percentage difference threshold set by the operator to save the quadrupole during the stacking process. |
| Lines 175 – 180 – "Both geo-electric and electromagnetic," Please use a consistency term to represent ERT. For example, "Both electrical and electromagnetic" | Corrected, thanks (L 177). |
| Lines 175 – 180 – In the Fig. 1C it is difficult to understand the discussed lines in this sentence "an initial areal FDEM acquisition was followed by 3 additional lines to intercept seeder heavy passages, and 8 detailed transects, both FDEM and ERT (4 along and 4 across normal tractor tramlines. Please modify the Fig. 1C for clarity. | We modified Fig. 1C as suggested by the other Reviewers and specified in the caption that it refers to the initial areal FDEM survey (L 228). As described, in Fig. 1 we can understand the arrangement of the 3 lines intercepting the seeders passage (the results of which are in Fig. 6), and that of the 8 detailed transects (the results of which are in Fig. 7 and Fig. 8). |
| Line 185 – "dij" should be "dih"? | Exactly, corrected (L 186). |
| Lines 185 – 190 – "For each area, one geophysical detailed survey (i.e. ERT + FDEM) was performed." Please rewrite this sentence for clarity. One geophysical detailed survey or both surveys (ERT + FDEM)? | Corrected, thanks (LL 190-191). |

| | |
|---|---|
| Lines 190 – 195 – "throughout the 0–80 cm soil layer." Please provide the soil sampling depth intervals. Same in " 0.70 m" as well. | Corrected, thanks (LL 195-196). |
| Lines 195 – 200 – "bulk density" should be "BD", "4m" should be "4 m" | Both corrected, thanks (LL 199-201). |
| Lines 200 – 205 – "22cm" should be "22 cm" | Corrected, thanks (L 204). |
| Heading 3.1 – What is mean by "First" here | Substituted with "Areal" to be clearer (L 215). |
| Lines 215 – 220 – Please introduce the abbreviations (VCP0.32 and HCP1.18) in the first use. | Already introduced in LL 128-129 |
| Lines 220 – 225 and Figure 2a - I'm just wondering why the legend of Fig 2a ranges only to 30 mS/m and not 40 mS/m? In the content authors mentioned "In the bottom layer (HCP1.18), a maximum increase of approximately 15 mS m-1 is observed, with values exceeding 40 mS m-1 in the most conductive zones," | We modified the colorscale range of Fig. 2 to be consistent with the other figures, as suggested. |
| Figure 2.a – Please rearrange the overlapped labels | Figure 2 has been modified and improved |
| Figure 3 – "a" should be removed. The sample symbol in the legend is different from the map. | Removed and improved caption, thanks (L 231). The "sample" symbol is a red dot, in each cluster 5 neighboring samples were collected (as shown in Fig. 1C) so it is just the effect of multiple close dots. |
| Lines 230 – 235 – "FDR" should be "TDR" | Corrected, thanks (L 234). |
| Figure 4 – VWC please introduce the abbreviation first. | Already introduced in L 198. |
| Lines 240 – 245, and 255 – 260 – Why the unit for water content in kg/kg. Please correct the unit of volumetric water content to m3 /m3 , as similar to the figure 5. | Thanks for the correction, fixed (LL 246-248) |
| Lines 320 -325 – Please remove the already introduced abbreviations. | Removed to be consistent in the whole paragraph, thanks. |
| Lines 340 -345 – "In both areal and detailed surveys, the highly compacted portions of the soil are characterized by high electrical conductivity anomalies relative to the context." What would be the reason for this observation, please discuss. | We previously introduced the relationship between water content and EC, with positive site-specific correlations (LL 340-346). We then expanded the text, discussing the reasons for our observation (LL 345-346). |
| Lines 375 – 380 – That would be nice if the authors could mention the most important challenges here. "Despite its potential and widespread application, the use of | Thanks for the suggestion, we improved the text (LL 383-384). |

| electromagnetic geophysics in agriculture presents challenges such as……" | |
|---|---|